# Multiwalled Carbon Nanotubes Promote Bacterial Conjugative Plasmid Transfer

Katrin Weise,[a] Lena Winter,[a] Emily Fischer,[a] David Kneis,[a] Magali de la Cruz Barron,[a,b] Steffen Kunze,[a] Thomas U. Berendonk,[a] Dirk Jungmann,[a] Uli Klümper[a]

[a]Technische Universität Dresden, Institute of Hydrobiology, Dresden, Germany
[b]Helmholtz Centre for Environmental Research GmbH - UFZ, Department of River Ecology, Magdeburg, Germany

**ABSTRACT** Multiwalled carbon nanotubes (MWCNTs) regularly enter aquatic environments due to their ubiquity in consumer products and engineering applications. However, the effects of MWCNT pollution on the environmental microbiome are poorly understood. Here, we evaluated whether these carbon nanoparticles can elevate the spread of antimicrobial resistance by promoting bacterial plasmid transfer, which has previously been observed for copper nanomaterials with antimicrobial properties as well as for microplastics. Through a combination of experimental liquid mating assays between *Pseudomonas putida* donor and recipient strains with plasmid pKJK5::*gfpmut3b* and mathematical modeling, we here demonstrate that the presence of MWCNTs leads to increased plasmid transfer rates in a concentration-dependent manner. The percentage of transconjugants per recipient significantly increased from $0.21 \pm 0.04\%$ in absence to $0.41 \pm 0.09\%$ at 10 mg $L^{-1}$ MWCNTs. Similar trends were observed when using an *Escherichia coli* donor hosting plasmid pB10. The identified mechanism underlying the observed dynamics was the agglomeration of MWCNTs. A significantly increased number of particles with $>6$ $\mu$m diameter was detected in the presence of MWCNTs, which can in turn provide novel surfaces for bacterial interactions between donor and recipient cells after colonization. Fluorescence microscopy confirmed that MWCNT agglomerates were indeed covered in biofilms that contained donor bacteria as well as elevated numbers of green fluorescent transconjugant cells containing the plasmid. Consequently, MWCNTs provide bacteria with novel surfaces for intense cell-to-cell interactions in biofilms and can promote bacterial plasmid transfer, hence potentially elevating the spread of antimicrobial resistance.

**IMPORTANCE** In recent decades, the use of carbon nanoparticles, especially multiwalled carbon nanotubes (MWCNTs), in a variety of products and engineering applications has been growing exponentially. As a result, MWCNT pollution into environmental compartments has been increasing. We here demonstrate that the exposure to MWCNTs can affect bacterial plasmid transfer rates in aquatic environments, an important process connected to the spread of antimicrobial resistance genes in microbial communities. This is mechanistically explained by the ability of MWCNTs to form bigger agglomerates, hence providing novel surfaces for bacterial interactions. Consequently, increasing pollution with MWCNTs has the potential to elevate the ongoing spread of antimicrobial resistance, a major threat to human health in the 21st century.

**KEYWORDS** plasmid, MWCNT, horizontal gene transfer, antimicrobial resistance, agglomeration, emerging pollutants, nanoparticles

Carbon nanotubes (CNTs), and especially multiwalled carbon nanotubes (MWCNTs), are described to be the stars among nanomaterials (1). They are widely used in textiles, drug delivery, and health care products (2–4). Due to their broad usage, MWCNTs reach aquatic environments through direct routes such as wastewater of industrial production sites, while indirect introduction occurs from products via sewage and landfill

Address correspondence to Uli Klümper, Uli.Kluemper@tu-dresden.de.

The authors declare no conflict of interest.

leachates (5). Consequently, CNTs have been detected as pollutants in surface waters with concentrations ranging from 0.001 to 0.8 ng L$^{-1}$ (6, 7). Still, environmental concentrations of nanomaterials in general and CNTs in specific are widely unknown in a majority of environments (8). This is due to difficulties in their detection due to their size and their material, pure carbon.

Once MWCNTs enter the environment, they can have a variety of effects on environmental microbiomes. They have, for example, been shown to possess antimicrobial and cytostatic activities against certain bacterial strains (9, 10) and algae (11–13). Yang et al. (14) found that MWCNTs with smaller diameters generally exhibited more severe toxicity to different bacteria, with toxic effects being strain dependent. Furthermore, the accumulation of CNTs on the surfaces of *Chlorella vulgaris* cell walls resulted in shading effects and the inhibition of photosynthetic activity due to lower light availability (15). In addition, increases in the activity of the antioxidant enzyme superoxide dismutase, one of the most important antioxidant enzymes for the protection of cells against reactive oxygen species (ROS), was detected when microorganisms were exposed to MWCNTs (15).

Recently, it has been reported that the presence of MWCNTs can also lead to an increase in antibiotic resistance genes (ARG) in activated sludge sequencing batch bioreactors (16), with the underlying mechanisms yet poorly understood. Assessing how exactly these common environmental pollutants affect the proliferation of ARGs remains an important task (17), as the spread of antimicrobial resistance (AMR) has become a global threat to human and environmental health (18). Here, we hypothesize that the presence of MWCNTs can directly promote plasmid-mediated, bacterial horizontal gene transfer rates, a key process in the spread of AMR within bacterial populations but also across highly diverse bacterial strains (19, 20).

This hypothesis is based on previous reports that the exposure to certain nano- and microparticles can indeed affect bacterial plasmid transfer. For example, copper nanoparticles were able to promote horizontal plasmid transfer at environmentally relevant concentrations mainly due to causing an overproduction in ROS in the exposed bacteria (21). This led to an upregulation of expression levels of genes and proteins related to oxidative stress, cell membrane permeability, and pilus generation. Similar effects have been described for copper ions (21) as well as non-antibiotic pharmaceuticals that are able to trigger the bacterial SOS stress response and increase cell wall permeability, hence resulting in increased plasmid uptake rates (22, 23). Such increases in the production of ROS have also been described for CNTs (24), suggesting that exposure to CNTs might also lead to elevated plasmid transfer rates. Further, particles without antibacterial properties, such as microplastics, are able to promote plasmid transfer due to providing novel surfaces for biofilm formation and hence donor-recipient interactions with intense cell-cell contact (25). Garacci et al. (26) and Weise et al. (27) demonstrated a type of network formation between graphene and MWCNTs with the extracellular polymeric substances of benthic organisms. Accordingly, it is also possible that bacteria colonize MWCNT agglomerates, which could facilitate biofilm formation or promote bacterial agglomeration (28), thus promoting horizontal gene transfer.

To test our hypothesis that MWCNTs affect bacterial plasmid transfer, we here combined experimental plasmid transfer assays with mathematical modeling of plasmid transfer dynamics. *Pseudomonas putida* KT2440 served as the bacterial plasmid donor and recipient strain and was exposed to different concentrations of MWCNTs. We further tested if either of the proposed mechanisms, overproduction of ROS or providing novel surfaces, could play a role in the effects of MWCNTs on bacterial plasmid transfer.

## RESULTS

**MWCNTs do not affect bacterial growth.** Within the 24 h of incubation, bacterial densities increased in the control without MWCNT from initially ~10$^6$ cells/mL to 3.09 ± 1.10 × 10$^8$ mL$^{-1}$ (*P. putida* donor of pKJK5) and 5.61 ± 1.02 × 10$^8$ mL$^{-1}$ (*P. putida* recipient), respectively (Fig. 1). The presence of MWCNTs in the microcosms

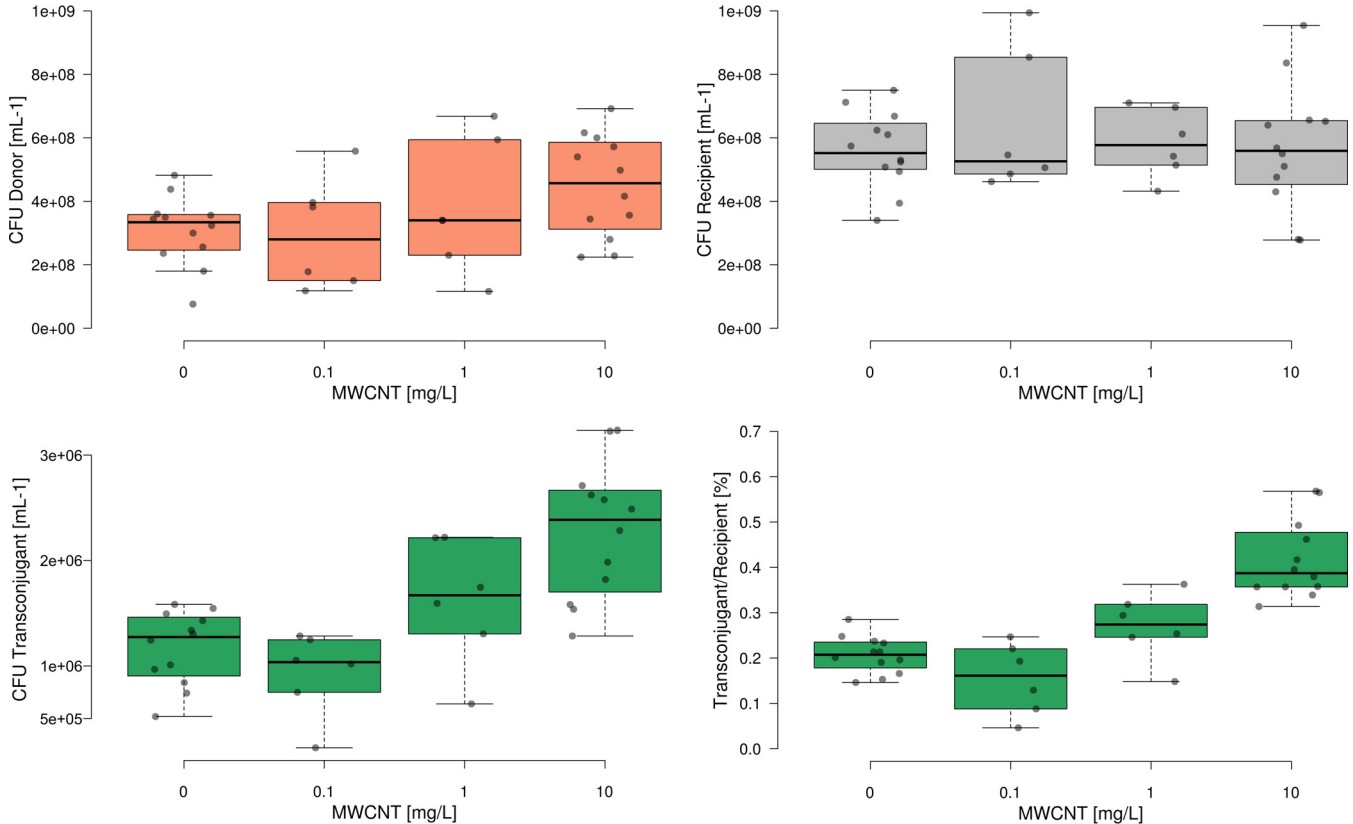

**FIG 1** Final concentrations of bacterial donor *P. putida* with plasmid pKJK5 (red), recipient strain *P. putida* (gray), transconjugants (green), and ratio of transconjugants per recipient (green) after incubation with various concentrations of MWCNTs (*n* = 12, 6, 6, 12). Center lines show the median; box limits indicate the 25th and 75th percentiles; whiskers extend 1.5 times the interquartile range from the 25th and 75th percentiles; outliers are represented by dots; data points are plotted as circles.

across all concentrations did not cause any significant effect on bacterial densities of either the *P. putida* donor or the recipient strain compared to the no-MWCNT control (all $P > 0.05$, $n = 6$ to 12, analysis of variance [ANOVA]).

Similar results were observed when using plasmid pB10, where exposure to MWCNTs at any concentration had equally no significant effect on bacterial densities of either the *Escherichia coli* donor or the *P. putida* recipient (all $P > 0.05$, $n = 3$, ANOVA) (Fig. 2). Consequently, MWCNTs do not affect bacterial growth under the conditions used in the experiments.

**MWCNTs promote plasmid transfer.** Despite the constant cell densities of donor and recipient strain, the absolute number of transconjugants receiving plasmid pKJK5 increased significantly with increasing concentrations of MWCNTs ($r_s = 0.66603$, $P < 0.001$ of Spearman's rho being zero; Fig. 1) from $1.17 \pm 0.34 \times 10^6$ mL$^{-1}$ in the absence of MWCNTs to $2.28 \pm 0.64 \times 10^6$ mL$^{-1}$ at 10 mg L$^{-1}$ MWCNTs. The concentration-response behavior held true when normalizing the absolute number of transconjugants to the number of recipient cells ($r_s = 0.75947$, $P < 0.001$ of Spearman's rho being zero; Fig. 1) where an approximately 2-fold increased likelihood of a recipient to become a transconjugant was observed. The percentage of transconjugants per recipient (T/R) increased from $0.21 \pm 0.04\%$ in absence to $0.41 \pm 0.09\%$ at the highest tested MWCNT concentration ($Q = 8.44$, $P < 0.001$, post hoc Tukey's honestly significant difference [HSD] test, Fig. 1).

To confirm that these trends hold true for different plasmid:donor combinations, experiments were repeated with plasmid pB10 introduced through an *E. coli* donor strain. Again, the absolute number of transconjugants receiving plasmid pB10 increased significantly with increasing concentrations of MWCNTs ($r_s = 0.82048$, $P = 0.001$ of Spearman's rho being zero; Fig. 2) from $1.37 \pm 0.55 \times 10^6$ mL$^{-1}$ in the absence of MWCNTs to $8.03 \pm 0.98 \times 10^6$ mL$^{-1}$ at 10 mg L$^{-1}$ MWCNTs. Equally, T/R significantly increased from $0.08 \pm 0.05\%$ at 0 mg L$^{-1}$ MWCNTs to $0.43 \pm 0.15\%$ at 10 mg L$^{-1}$ MWCNTs ($t = -3.966$, $P = 0.017$, $n = 3$, $t$ test, Fig. 2).

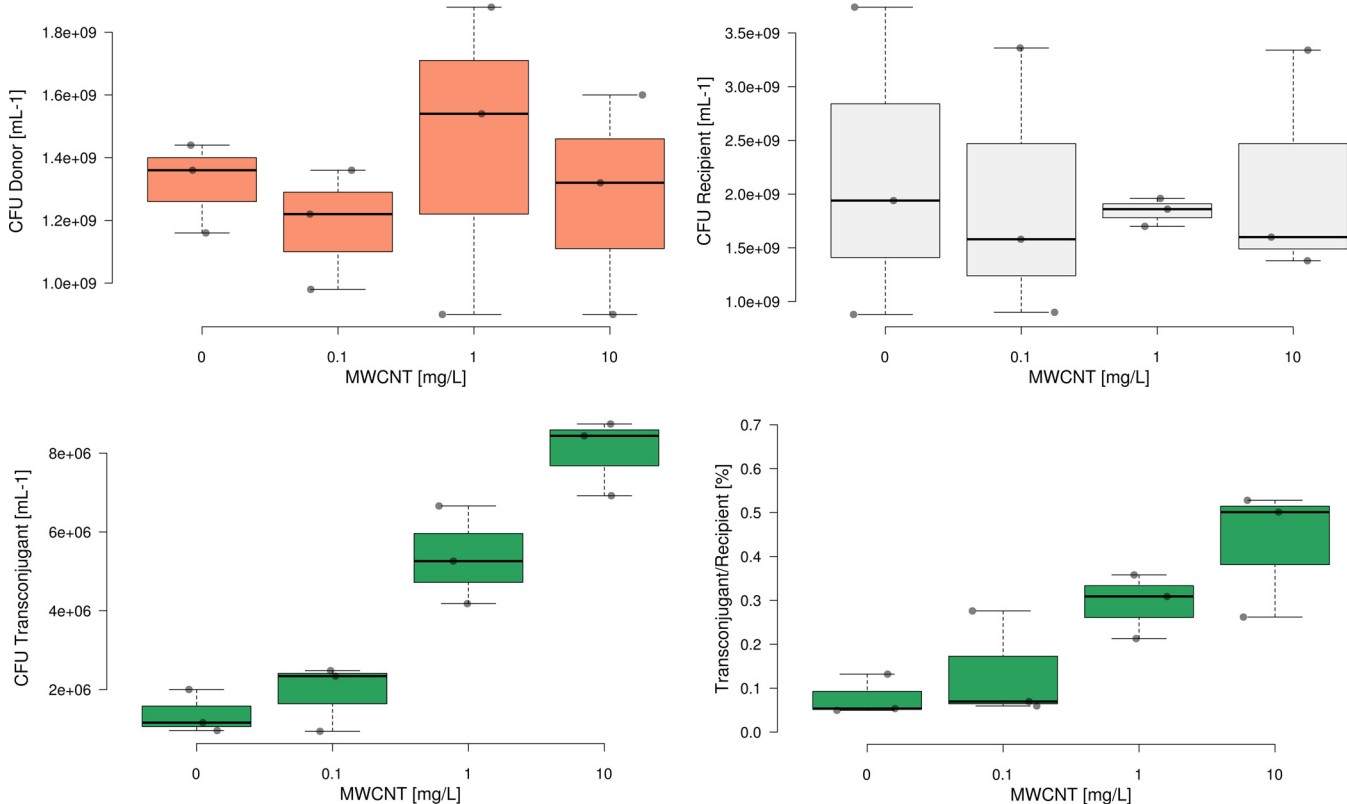

**FIG 2** Final concentrations of bacterial donor *E. coli* with plasmid pB10 (red), recipient strain *P. putida* (gray), transconjugants (green), and ratio of transconjugants per recipient (green) after incubation with various concentrations of MWCNTs (*n* = 3, 3, 3, 3). Center lines show the median; box limits indicate the 25th and 75th percentiles; whiskers extend 1.5 times the interquartile range from the 25th and 75th percentiles; outliers are represented by dots; data points are plotted as circles.

As the observed dynamics were nearly identical for the different donor:plasmid combinations, subsequent analysis regarding the underlying mechanism was carried out exclusively for the *P. putida* donor and plasmid pKJK5.

**Modeling the dependency of plasmid transfer on MWCNT exposure.** To determine the nature of the relationship between MWCNT exposure and plasmid transfer (e.g., a direct concentration dependence), we employed modeling of the plasmid transfer dynamics of plasmid pKJK5 between the *P. putida* donor and recipient. Given independent estimates of the carrying capacity ($\kappa = 1 \times 10^9$ cells mL$^{-1}$) and the growth rate constant ($\mu = 1$ h$^{-1}$), fitting of the ordinary differential equation (ODE) model of mating experiments (see Material & Methods) to the observed data at $T_{24h}$ yielded estimates for all parameters of equation 1 describing the dependency of the plasmid transfer rate on MWCNT concentrations:

$$f = 10^{(-a + b \times X^c)} \tag{1}$$

where the empirical parameter "*a*" represents the basal plasmid transfer efficiency in the absence of MWCNT while "*b*" and "*c*" describe a possible exposure effect in the form of a power law. This includes the basal transfer efficiency (*a* = 12.6) and the coefficients reflecting the effect of MWCNT exposure (*b* = 0.0922, *c* = 0.557). The value of *b* > 0 confirms a positive relationship between MWCNT exposure and plasmid transfer within the range of conditions tested. The value of *c* < 1 clearly suggests a non-linear dependency where an ever-increasing exposure to MWCNTs does not stimulate plasmid transfer further. Comparing predictions of the fitted model to observations reveals notable mismatches for individual experimental runs (Fig. 3). However, residuals are well-behaved in the sense that their values are independent of the magnitude of the

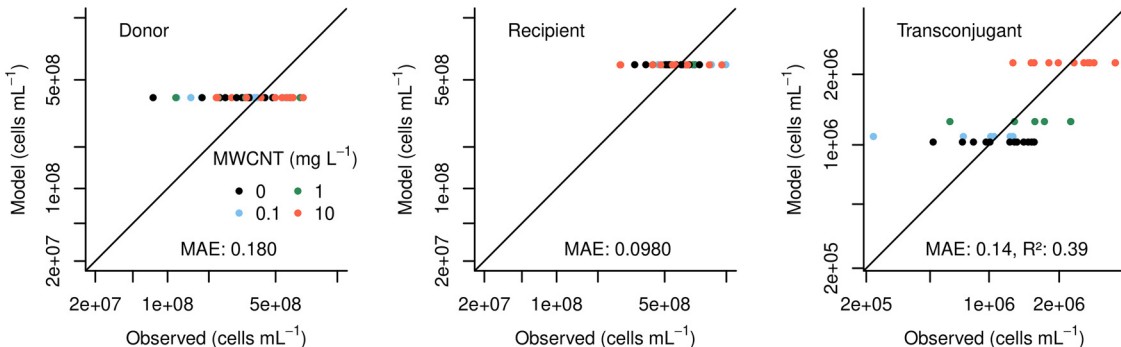

FIG 3 Predictions of the fitted plasmid transfer model (equations 1 to 4) assuming an immediate effect of MWCNTs on the plasmid transfer rate in comparison to the experimental observations at $T_{24h}$ for the three state variables: donor, recipient, and transconjugant. For a perfect fit, all dots would appear on the diagonal (MAE, mean absolute error in cells/mL [in $\log_{10}$ units]; $R^2$, fraction of explained variance).

fitted variable and their distribution is consistent with a zero-mean normal distribution. The mean absolute errors (MAE) of less than 0.2 log units are within the precision limits of bacterial enumeration via plating.

**ROS scavenging does not affect plasmid transfer across MWCNT gradient.** To gain mechanistic insights into whether a potential overproduction of ROS in response to exposure to MWCNTs could explain the observed increase in plasmid transfer in *P. putida*, we repeated the plasmid transfer experiment for the control and highest MWCNT concentration in the presence of 100 $\mu$mol thiourea, a well-known ROS scavenger. At the selected concentration, thiourea did not affect growth of the donor and recipient bacteria. Again, significant increases in plasmid transfer (transconjugants per recipient) were observed in the presence of MWCNTs. In the absence of thiourea, plasmid transfer increased from 0.21 $\pm$ 0.03% (0 mg L$^{-1}$ MWCNT) to 0.38 $\pm$ 0.05% (10 mg L$^{-1}$ MWCNT) in a significant manner ($P = 0.000021$, $n = 6$, $F = 55.83$, ANOVA; Fig. 4). In the presence of thiourea, a similar effect was observed, with plasmid transfer increasing from 0.21 $\pm$ 0.05% to 0.46 $\pm$ 0.10% ($P = 0.00034$, $n = 6$, $F = 28.21$). However, both in the absence ($P = 0.970$, $n = 6$, $F = 0.0015$) and in the presence of 10 mg L$^{-1}$ MWCNTs ($P = 0.121$, $n = 6$, $F = 2.87$), no significant effect of ROS scavenging thiourea on plasmid transfer was observed. Experiments were repeated with a 100-fold increased concentration of thiourea (10 mM), with observed results being identical (data not shown). Consequently, the overproduction of ROS in response to the exposure to MWCNTs plays a negligible role, if any, in promoting plasmid transfer.

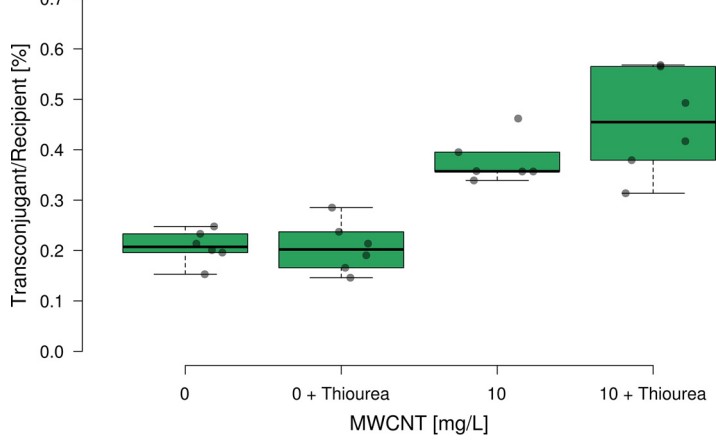

FIG 4 Ratio of transconjugants per recipient after incubation with various concentrations of MWCNTs in the presence and absence of the ROS scavenger thiourea at a concentration of 100 $\mu$mol ($n = 6$). Center lines show the median; box limits indicate the 25th and 75th percentiles; whiskers extend 1.5 times the interquartile range from the 25th and 75th percentiles; outliers are represented by dots; data points are plotted as circles.

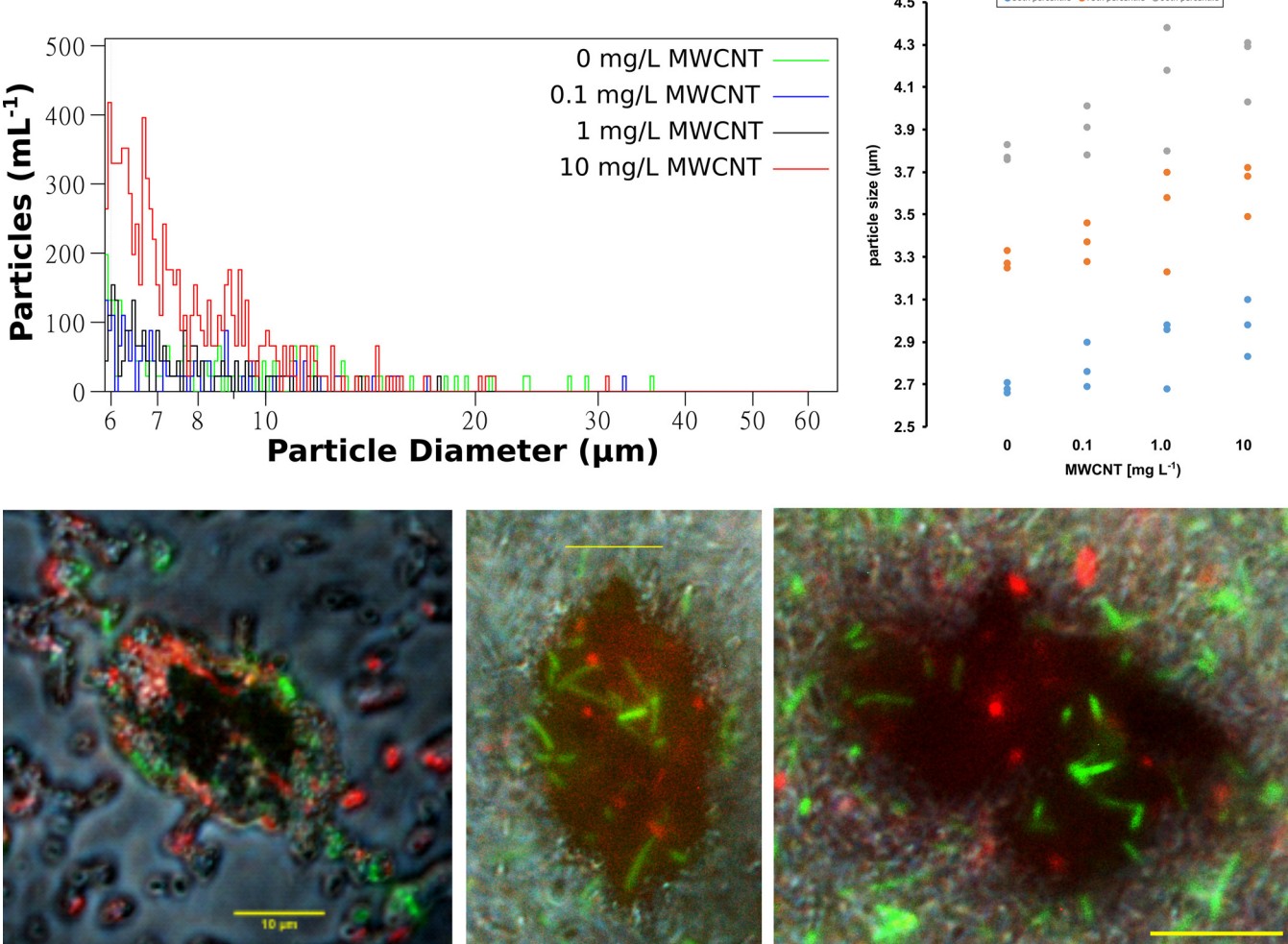

**FIG 5** Agglomeration of MWCNTs and their colonization. Top left: particle size distribution per milliliter in microcosms with different MWCNT concentrations. Concentration of particles with a diameter between 6 and 60 $\mu$m, large enough to be colonized by a biofilm. Top right: average particle size in the microcosms at the 50th, 75th, and 90th percentiles. Bottom: fluorescent microscopic images of MWCNT agglomerates with attached biofilm of *Pseudomonas putida* including donor bacteria (red) and transconjugants (green).

**Increased agglomeration through MWCNTs promotes plasmid transfer.** MWCNTs are known for their ability to form agglomerates when in suspension. Agglomerates could provide novel surfaces for colonization and hence promote plasmid transfer in biofilms with increased cell-to-cell interactions. To test if this is the case, the particle size distribution within the microcosms across the gradient of MWCNT concentrations was measured. In the presence of the highest concentrations of MWCNTs, indeed a more than 3-fold higher number of particles per milliliter with a diameter between 6 and 60 $\mu$m, large enough to be MWCNT-bacteria agglomerates, was detected compared to the control (Fig. 5, top left). Overall, the particle size at the 50th, the 75th, and the 90th percentiles increased significantly with increasing MWCNT concentrations ($p_{50} = 0.038$, $R_{50} = 0.603$; $p_{75} = 0.027$, $R_{75} = 0.633$; $p_{90} = 0.048$, $R_{90} = 0.581$; Pearson correlation, $n = 12$) (Fig. 5, top right). To test whether these particles have been colonized by bacterial biofilm, images of particles were taken using fluorescence microscopy. Three channels were combined: the brightfield to detect agglomerated MWCNTs and bacteria, red fluorescence to detect donor bacteria, and green fluorescence to detect transconjugants. MWCNT agglomerates (in dark black) were covered in bacterial biofilm that contained donor bacteria as well as elevated amounts of green transconjugant cells (Fig. 5, bottom). Consequently, MWCNT agglomerates can serve as a novel surface for intense cell-to-cell interactions in biofilms and can, hence, promote bacterial plasmid transfer in the microcosms.

## DISCUSSION

Here, we demonstrate that the presence of MWCNTs can have a significant effect on bacterial plasmid transfer in liquid-phase environments, leading to a more than 2-fold increase of observed transconjugants per recipient. This effect was demonstrated for two different plasmid:donor combinations. An MWCNT concentration dependence of the observed increase in plasmid transfer was detected through statistical analysis as well as modeling of plasmid transfer dynamics. The main identified mechanism underlying the observed dynamics was the agglomeration of MWCNTs, leading to a significantly increased number of particles with an $>6~\mu$m diameter which can in turn be colonized by bacterial biofilms. This ability to form MWCNT-bacterial agglomerates has been described previously (28). In biofilms, bacteria experience more intense bacterial cell-to-cell contact and a higher probability of interactions. Hence, niches with increased plasmid transfer rates between neighboring cells can be formed (29). Similar effects have been observed for microplastics that can lead to elevated plasmid transfer rates in aquatic environments by providing a novel colonization surface (25).

As a second potential mechanism, we investigated if the ability of MWCNTs to induce the bacterial SOS stress response by causing elevated levels of ROS in bacteria (24) could promote plasmid transfer. This mechanism triggering elevated bacterial plasmid transfer rates was previously demonstrated for copper nanoparticles (21) as well as non-antibiotic pharmaceuticals (22). However, in the presence of thiourea, a commonly used ROS scavenger (22, 23), plasmid transfer rates remained elevated in the presence of MWCNTs, indicating that, at least under the nutrient-rich conditions used here, the effect of MWCNTs to trigger bacterial stress remains negligible. This is further supported by the fact that no antibacterial effect of MWCNTs causing lower bacterial numbers in the microcosms was observed, despite the highest used concentration of 10 mg L$^{-1}$ being above previously reported inhibitory concentrations for CNTs (24). This effect is potentially owed to the larger size of MWCNTs compared to that of single walled CNTs, which decreases these properties (14, 24).

Further, special functionalized MWCNTs loaded with plasmid DNA could enhance plasmid introduction through transformation into various bacterial hosts as well as *Ctenopharyngodon idellus* kidney (CIK) cells and are hence biotechnologically exploitable as a novel gene transfer vector system (30–32). These functionalized MWCNTs tend to align their tips perpendicularly to the cell surface and appear to "prick" the cells like a needle, leading to higher numbers of transformed cells that express the plasmid-carried genes than numbers of cells that were treated with naked DNA (30). However, in this previous study, high concentrations (0.3 $\mu$g mL$^{-1}$) of specifically extracted plasmid DNA were necessary to observe effects. On the contrary, in our experimental design, as well as in most environmental settings, concentrations of naked plasmid DNA, if present due to bacterial lysis in the culture, would be expected to be several orders of magnitude lower. Furthermore, the association of this free DNA with the here-used, non-functionalized MWCNTs is highly unlikely. Consequently, this mechanism, while biotechnologically exploitable, will most likely not have contributed to the observed results.

To assess if the dynamics observed in this study constitute a risk associated with MWCNT pollution of aquatic environments, usually the risk quotient, calculated from the predicted environmental concentration (PEC) and the extrapolated effect concentrations (predicted no effect concentration, PNEC), is assessed. If this quotient is larger than 1, a risk can be assumed (33). The effect concentration was here determined as 10 mg L$^{-1}$ of MWCNTs at which a constant and significant effect on plasmid transfer was observed. However, there are only very limited measured environmental concentrations for CNTs, as they consist of pure carbon and can hardly be distinguished from other carbon in the environment. Still, the few known aquatic environmental concentrations remain significantly below the here-determined effect concentration and are usually reported in the ng L$^{-1}$ range (6, 7). A further consideration to make is the mode of effect. While providing novel surfaces for colonization through agglomeration of MWCNTs might play a role in this artificial experimental system, the number of

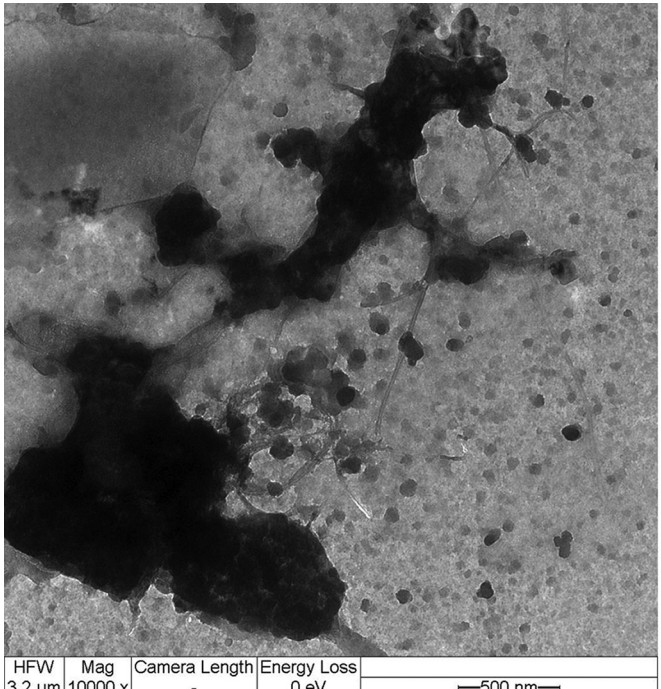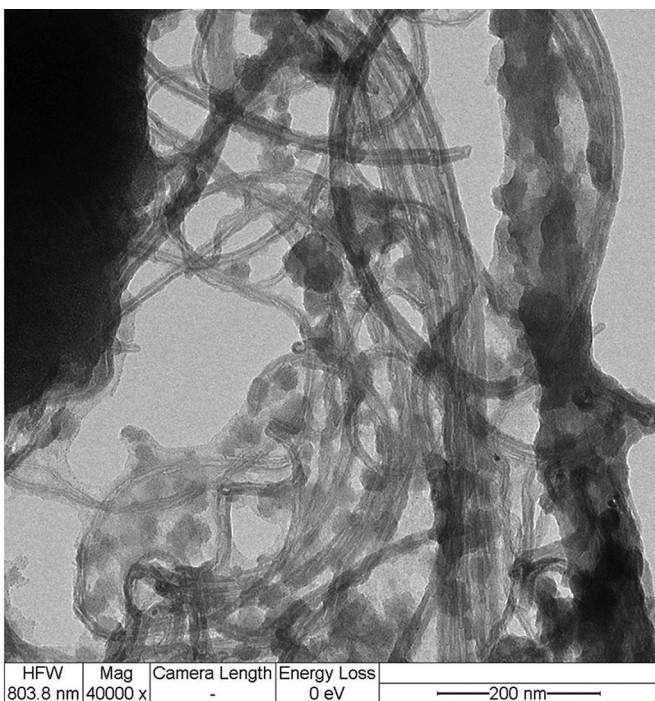

**FIG 6** TEM image of MWCNT agglomerates (black cluster, left) and zoom into single strands (right).

natural particles will in most environments far outweigh that of particles created through newly introduced MWCNT pollution. However, a previous study on microplastic particles has proven that these anthropogenically introduced surfaces provide a potential to promote gene transfer far greater than that of naturally occurring particles (25). Further, MWCNTs have special sorption properties (34, 35). Depending on pH, temperature, and redox processes, CNTs are able to adsorb hydrophobic environmental chemicals such as antibiotics (36) or heavy metal cations (37). This has the potential to result in agglomerates of MWCNTs that not only promote plasmid transfer of ARGs but are also enriched in agents well known to select for these ARGs, hence resulting in hot spots of AMR proliferation. Aside from the environment, the reported results could pose relevant for the medical application of MWCNTs. CNTs, with their ability of high adhesion to bacterial cells (38), have for example been suggested as useful biomaterials for the elimination of human oral pathogens (39). However, creating CNT-bacterial agglomerates might have detrimental effects, as it could lead to increased gene transfer between human-pathogenic bacterial strains and the potential formation of multiresistant superbugs.

**Conclusion.** We here demonstrate that MWCNTs are indeed able to promote bacterial plasmid transfer by providing novel surfaces for bacterial interactions through agglomeration. While a pronounced effect was observed under laboratory conditions, the identified effect concentrations as well as the effect mode suggest that MWCNT pollution provides only a minor risk to significantly enhance the proliferation of AMR in the environment. Our results provide concise and important insights into the effects of MWCNTs on microorganisms that should be regarded when considering the use of MWCNTs for medical application.

## MATERIALS AND METHODS

**Multiwalled carbon nanotubes.** The applied pristine MWCNTs (Baytubes C 150 P, BTS, Germany) were purchased from Bayer Material Science AG 2007 and visualized via transmission electron microscopy (TEM). Properties of the used MWCNTs [Baytubes, purity of >95%] are as follows: number of walls: 3 to 15; outer diameter distribution: 5 to 20 nm; inner diameter distribution: 2 to 6 nm; length: 1 to 10 $\mu$m; bulk density: 140 to 160 kg/m$^3$.

A detailed description and characterization of MWCNTs can be found in Politowski et al. (40, 41) and Weise et al. (27). Agglomerates with their typical cross-link structure and single strands of MWCNTs displaying a spiral shape with some irregularities were identified (Fig. 6). A stock solution with a concentration of

100 mg L$^{-1}$ pristine MWCNTs was freshly prepared. For this, 50 mg of sterile MWCNT agglomerates were transferred into a glass beaker with 500 mL sterile deionized H$_2$O. Afterwards, dispersion by an ultrasonic probe (Sonopuls HD 2070, 70 W, pulse: 0.2 s, pause: 0.8 s, Bandelin, Germany) was applied for 45 min to ensure single MWCNT strands. After dispersion, 0, 0.025 mL, 0.25 mL, and 2.5 mL from the MWCNT stock solution were withdrawn, filled up to a final volume of 2.5 mL with sterile deionized H$_2$O, and added to the microcosms, resulting in final MWCNT concentrations of 0, 0.1, 1, and 10 mg L$^{-1}$.

**Bacterial strains and culture conditions.** *Pseudomonas putida* KT2440 was used as a plasmid donor and a plasmid recipient strain in liquid mating assays. For the recipient strain, a spontaneous rifampicin-resistant mutant of the nonfluorescent *Pseudomonas putida* KT2440 wild-type strain (42) was created. A total of 100 $\mu$L of an overnight culture of the wild-type strain was plated on LB (10 g L$^{-1}$ tryptone, 5 g L$^{-1}$ yeast extract, 5 g L$^{-1}$ NaCl) agar plates containing 200 $\mu$g mL$^{-1}$ rifampicin. After cultivation at 37°C overnight, a rifampicin-resistant mutant colony was picked and grown in LB liquid medium containing 200 $\mu$g mL$^{-1}$ rifampicin and served subsequently as the recipient strain.

*Pseudomonas putida* KT2440::*lacI$^q$-pLpp-mCherry-Km$^R$* (43) carrying the IncP-1$\epsilon$ broad-host-range plasmid pKJK5::*gfpmut3b* (44) was used as the donor strain. Plasmid pKJK5 encodes tetracycline and trimethoprim resistance and has a very broad transfer range to both Gram-negative and Gram-positive phyla (19). The plasmid was marked with an entranceposon-derived (45) genetic tag that carries a *lacI$^q$* repressible promoter upstream the conditionally expressed *gfpmut3b* gene, encoding green fluorescent protein (GFP). The donor strain was chromosomally tagged with a gene cassette encoding kanamycin resistance as well as constitutive red fluorescence, expressed by the *mCherry* gene, and constitutive *lacI$^q$* production, repressing the plasmid's *lacI$^q$* promoter. As a result, *gfp* expression is repressed in the donor strain but expressed in recipients upon successful receipt of the plasmid, resulting in green fluorescent transconjugant cells. Hence, plasmid transfer can be confirmed by fluorescence microscopy (46). Donor and recipient bacteria were grown overnight in LB medium at 30°C at 150 rpm and supplemented with either tetracycline (10 $\mu$g mL$^{-1}$) or rifampicin (100 $\mu$g mL$^{-1}$), respectively, harvested by centrifugation, washed in 0.9% sterile NaCl solution, adjusted in cell density (OD$_{600}$ = 1), and used in liquid mating assays.

**Plasmid transfer assay.** To quantify plasmid transfer across a gradient of MWCNT concentrations, liquid mating assays between the donor and the recipient strain were carried out in microcosms. Mating experiments were performed in glass vials containing 25 mL sterile, liquid LB medium supplemented with appropriate concentrations of MWCNTs (0, 0.1, 1, 10 mg L$^{-1}$) at 3 to 12 replicates per concentration. Vials were inoculated with 500 $\mu$L of optical density (OD)-adjusted donor and recipient strain solution resulting in initial concentrations of approximately 10$^6$ cells mL$^{-1}$ each of the donor and recipient strains. The mating mixture was incubated at 30°C under continuous shaking at 180 rpm.

After 24 h, samples were taken from each replicate microcosm, serially diluted, plated on LB selective plates with 50 $\mu$g mL$^{-1}$ kanamycin and 10 $\mu$g mL$^{-1}$ tetracycline (donor), 100 $\mu$g mL$^{-1}$ rifampicin (recipient), or 50 $\mu$g mL$^{-1}$ rifampicin and 5 $\mu$g mL$^{-1}$ tetracycline (transconjugants), and subsequently incubated (48 h, 30°C). Thereafter, colonies were counted. Similarly, the initial donor and recipient cell densities were quantified immediately after inoculation of the microcosms. Note that the selective concentrations for the combination of rifampicin and tetracycline had to be lowered, as those two antibiotics have a synergistic effect in inhibiting growth of *P. putida*. Further, when quantifying recipients and transconjugants on rifampicin-containing plates, colonies were checked under a fluorescence microscope for red fluorescence to detect any potentially occurring spontaneous rifampicin mutants of the donor strain and for green fluorescence to confirm the successful receipt of plasmid pKJK5::*gfpmut3b*.

To confirm if observed results are exclusive to the above donor-recipient combination or to the pKJK5 plasmid, experiments were repeated using *Escherichia coli* MG1655::*lacI$^q$-pLpp-mCherry-Km$^R$* (43) as the donor strain of IncP-1$\beta$ broad-host range plasmid pB10 (47, 48) that confers resistance to amoxicillin, tetracycline, sulfonamides, and streptomycin. Conditions were exactly as described above with the lone exception that the incubation temperature was increased to 37°C.

**Plasmid transfer model.** To quantify the impact of MWCNT on the efficiency of plasmid transfer in a mechanistic framework, we employed a model of ordinary differential equations (ODE). The latter equation captures the dynamics of recipient (*R*), donor (*D*), and transconjugant (*T*) populations in liquid cultures due to the simultaneous effects of growth and conjugation (49, 50). Such models can readily be adapted to account for the impact of various environmental conditions on plasmid transfer efficiencies (48). Here, we amend the equations of Levin et al. (49) with an empirical term reflecting the effect of MWCNT on the rate constant of plasmid transfer. In contrast to empirical measures of plasmid transfer efficiency, such as T/R ratios, such model-based rate constants are more specific and better suited for comparison across different studies (50). The simultaneous ODE are given by equation 2:

$$\begin{bmatrix} \dfrac{d}{dt}D \;=\; \alpha \,\times\, D \\[2mm] \dfrac{d}{dt}R \;=\; \alpha \,\times\, R - \beta \\[2mm] \dfrac{d}{dt}T \;=\; \alpha \,\times\, T + \beta \end{bmatrix} \tag{2}$$

where the terms "$\alpha$" and "$\beta$" implement growth and plasmid-based horizontal gene transfer, respectively. Specifically, it is assumed that bacteria exploit the medium according to its carrying capacity ($\kappa$) and that a single intrinsic growth rate constant ($\mu$) is representative of all three bacterial strains (equation 3). Importantly, $\mu$ is independent of MWCNT levels in accordance with experimental findings (see "MWCNTs do not affect bacterial growth").

$$\alpha = \mu \times \left(1 - \frac{D + R + T}{\kappa}\right) \tag{3}$$

The gene transfer term $\beta$ (equation 4) represents the probability of recipients ($R$) meeting a plasmid-carrying partner ($D$, $T$) multiplied with a small positive rate constant ($f$) carrying the unit (cells mL$^{-1}$)$^{-1}$ h$^{-1}$. Here, the rate constant is dependent on the MWCNT concentration ($X$) according to equation 1:

$$\beta = f(X) \times R \times (D + T) \tag{4}$$

$$f = 10^{(-a + b \times X^c)} \tag{5}$$

where the empirical parameter "$a$" represents the basal plasmid transfer efficiency in the absence of MWCNT while "$b$" and "$c$" describe a possible exposure effect in the form of a power law. Modeling was performed in R version 4.1 using the packages *rodeo* (51) for implementing the ODE, *deSolve* (52) for numerical integration, and *minpack.lm* (53) for parameter fitting.

**Evaluating the effect of reactive oxygen species.** To investigate whether reactive oxygen species (ROS) production induced by MWCNTs affects plasmid transfer rates, liquid matings at MWCNT concentrations of 0 and 10 mg L$^{-1}$ were carried out according to the protocol described above. Here, the medium was supplemented with the ROS scavenger thiourea (54) at a final concentration of 100 $\mu$mol L$^{-1}$ (according to reference 22).

**Particle size distribution.** Particle size distribution in the cultures was measured with a Coulter Counter Multisizer3 (CCM3; Beckman Coulter Inc., Miami, FL) after 24 h of incubation. ISOTON II was used as the electrolyte. The aperture diameter was 100 $\mu$m with an aperture current of 1,600 $\mu$A. Measurements were carried out in 3 replicates of 500 $\mu$L across 300 size bins ranging from 2 $\mu$m to 60 $\mu$m. The total particle size distributions across those bins were recorded. To identify colonizable agglomerates, the numbers of particles with a diameter of at least 6 $\mu$m were further assessed.

**Fluorescence microscopy.** To visualize bacteria-MWCNT agglomerates, associated biofilm bacteria, and successful plasmid transfer, a 10-$\mu$L drop of bacterial-MWCNTs culture was placed on a microscope cover slide. Images were taken using an epifluorescence inverted microscope (Nikon Eclipse Ti), with a 100$\times$ oil-immersion objective, equipped with a GFP and TRITC/CY3 filter cubes, and a DS-Fi1c digital camera (Nikon instruments). The sample was observed by first focusing on the MWCNT agglomerate in bright field mode. The agglomerates were then examined for mCherry (donor)- and GFP (transconjugant)-positive cells. Images in all three channels were captured from identical locations using the NSI-elements AR 5.11.01 software. For optimized visualization using the ImageJ software (version 1.53f51), the images were separately adjusted in brightness and contrast with additionally performed background subtraction. Furthermore, the images of all three channels were merged into one overall image using the operation Image Calculator so that both the agglomerate and the red and green fluorescent bacteria could be visualized.

## ACKNOWLEDGMENTS

This work was supported by the European Union's Horizon 2020 research and innovation program under the PRIMA program supported by the European Union under grant agreement no. 1822, the ANTIVERSA project, and the JPI AMR - EMBARK project funded by the Bundesministerium für Bildung und Forschung under grant numbers 01LC1904A and F01KI1909A. K.W. was supported through a reentry scholarship by the Graduate Academy of TU Dresden funded by the budget passed by the Saxon state parliament. Responsibility for the information and views expressed in the manuscript lies entirely with the author(s). We acknowledge Petr Formánek of Leibniz-Institute for Polymer Research Dresden e.V. (IPF) for part of the presented TEM pictures of pristine MWCNTs. We declare no competing interests.

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
