## [Reviewer comments · Microbiology Spectrum]

Microbiology Spectrum

Multi-walled carbon nanotubes promote bacterial conjugative plasmid transfer

Katrin Weise, Lena Winter, Emily Fischer, David Kneis, Magali de la Cruz Barron, Steffen Kunze, Thomas Ulrich Berendonk, Dirk Jungmann, and Uli Klümper

Corresponding Author(s): Uli Klümper, TU Dresden

Review Timeline:

Submission Date:	February 2, 2022
Editorial Decision:	March 5, 2022
Revision Received:	March 15, 2022
Accepted:	March 21, 2022

Editor: Zhenjiang Xu

Reviewer(s): The reviewers have opted to remain anonymous.

Transaction Report:

DOI: <https://doi.org/10.1128/spectrum.00410-22>

March 5, 2022

Dr. Uli Kluemper
Technische Universität Dresden, Institute of Hydrobiology
Dresden
Germany

Re: Spectrum00410-22 (Multi-walled carbon nanotubes promote bacterial conjugative plasmid transfer)

Dear Dr. Uli Kluemper:

Link Not Available

Sincerely,

Zhenjiang Xu

Journals Department
Reviewer comments:

Reviewer #1 (Comments for the Author):

The manuscript describes the analysis of the influence of multi-walled carbon nanotubes (MWCNTs) frequently found in aquatic environments due to entry through diverse consumer products on conjugative plasmid transfer. By performing standard mating assays and by mathematical modelling the authors showed that the presence of MWCNTs can increase conjugative plasmid transfer in a concentration-dependent manner. However, the concentrations which increase transfer frequencies by up to two-fold are markedly higher than those encountered in the environment. The authors discussed this issue critically in the manuscript.

The conclusions are justified by the results. The paper is generally well written. However, there are some sentences/statements

which should be rephrased to improve understandability.

Specific comments:

- l. 28: per recipient close to doubled from 0.21 {plus minus} 0.04 % in absence: Rephrase
- l. 148: serially diluted
- l. 157: to detect instead of deduct?
- l. 174: does not take a dependency: Please rephrase
- l. 181: takes a dependence on: Rephrase
- l. 221: did not cause any significant effect
- l. 232: increased likelihood
- l. 245: "residuals are well-behaved": What do you mean by this? Please rephrase
- l. 313: perpendicularly
- l. 338-339: for the potentially transferred antibiotic resistance genes: Please rephrase; this statement is not clear.

Reviewer #2 (Comments for the Author):

Comments to the manuscript: "Multi-walled carbon nanotubes promote bacterial conjugative plasmid transfer" by Weise et al. (Spectrum00410-22).

1. Weise et al. compared the conjugative plasmid transfer in the absence and/or presence of multi-walled carbon nanotubes (MWCNTs) in liquid mating assays. They showed that the presence of MWCNTs could promote the transfer of plasmid. The authors concluded that the agglomeration of MWCNTs increased the plasmid transfer rates. The effects of reactive oxygen species (ROS) caused by the presence of MWCNTs might not be large for the plasmid transfer. This study contained interesting aspects, but some parts of the manuscript was not necessarily clear to this reviewer.
2. They used IncP-1epsilon plasmid, pKJK5 as a model plasmid. This reviewer could understand that the authors always used this plasmid as a model, but was the promotion of plasmid transfer commonly observed in other plasmids? Did the authors have any comments on it?
3. As for the ROS, did they compare the amount of ROS in the absence and presence of MWCNT?
4. Figure 3. The y-axis of the left graph should be shown in exponential representation. Did the authors have any comments that the CFU of transconjugants and rate of transconjugant decreased in the presence of 0.1 mg/L MWCNT?
5. Lines 159-190, 236-247 and Figure 4. This reviewer could not understand the meaning why they introduced the modeling of plasmid transfer on MWCNT exposure. It might be helpful if the authors described why they calculated the rates, what they expected, and what the conclusion was based on the comparisons between the modeling and observations.
6. Line 253-253. The data were not shown. Please describe more clearly here, i.e., "(data not shown)".
7. Lines 275-280 and Figure 6. They just showed one figure here, and the figure might be a representative one in the microscopy images. Was it difficult to show more examples? How frequently did the observed such images?
8. Lines 315-316. It was not necessarily clear why the authors considered the naked plasmid DNA in their experiments were negligibly low. It might be better to show the amount of naked plasmid DNA in the past report in ref. #44.

Staff Comments:

Preparing Revision Guidelines

Please return the manuscript within 60 days; if you cannot complete the modification within this time period, please contact me. If you do not wish to modify the manuscript and prefer to submit it to another journal, please notify me of your decision immediately so that the manuscript may be formally withdrawn from consideration by Microbiology Spectrum.

Dear Editor,

Please find below our detailed responses to the reviewer comments. The original comments are shown in black. Our responses are shown in blue. New additions to the revised version of the manuscript are cited in red.

Uli Klümper

Reviewer comments:

Reviewer #1 (Comments for the Author):

The manuscript describes the analysis of the influence of multi-walled carbon nanotubes (MWCNTs) frequently found in aquatic environments due to entry through diverse consumer products on conjugative plasmid transfer. By performing standard mating assays and by mathematical modelling the authors showed that the presence of MWCNTs can increase conjugative plasmid transfer in a concentration-dependent manner. However, the concentrations which increase transfer frequencies by up to two-fold are markedly higher than those encountered in the environment. The authors discussed this issue critically in the manuscript.

The conclusions are justified by the results. The paper is generally well written. However, there are some sentences/statements which should be rephrased to improve understandability.

We thank the reviewer for their positive assessment of our study and have improved the manuscript according to the reviewer's comments.

Specific comments:

I. 28: per recipient close to doubled from 0.21 {plus minus} 0.04 % in absence: Rephrase

We have modified the sentence to now read:

“The percentage of transconjugants per recipient significantly increased from 0.21 ± 0.04 % in absence to 0.41 ± 0.09 % at 10 mg L^{-1} MWCNTs.”

I. 148: serially diluted

Has been corrected.

I. 157: to detect instead of deduct?

Has been corrected.

I. 174: does not take a dependency: Please rephrase

We have modified the sentence to now read:

“Importantly, μ is independent of MWCNT levels in accordance with experimental findings (see Results 3.1).”

I. 181: takes a dependence on: Rephrase

We have modified the sentence to now read:

“Here, the rate constant is dependent on the MWCNT concentration (X) according to Eq. 4”

I. 221: did not cause any significant effect

Has been modified.

I. 232: increased likelihood

Has been corrected.

I. 245: "residuals are well-behaved": What do you mean by this? Please rephrase

"Well-behaved" meant that (1) there is no indication for a dependency of residuals on the magnitude of the predicted variable and (2) the distribution of residuals is close to symmetric around zero without notable outliers. We have now qualified this expression in the manuscript:

"However, residuals are well-behaved in the sense that their values are independent of the magnitude of the fitted variable and their distribution is consistent with a zero-mean normal distribution. The mean absolute errors (MAE) of less than 0.2 log units are within the precision limits of bacterial enumeration via plating."

I. 313: perpendicularly

Has been corrected.

I. 338-339: for the potentially transferred antibiotic resistance genes: Please rephrase; this statement is not clear.

We have modified the sentence to increase the clarity. It now reads:

“This has the potential to result in agglomerates of MWCNTs that not only promote plasmid transfer of ARGs but are also enriched in agents well known to select for these ARGs, hence resulting in hot-spots of AMR proliferation.”

Reviewer #2 (Comments for the Author):

Comments to the manuscript: "Multi-walled carbon nanotubes promote bacterial conjugative plasmid transfer" by Weise et al. (Spectrum00410-22).

1. Weise et al. compared the conjugative plasmid transfer in the absence and/or presence of multi-walled carbon nanotubes (MWCNTs) in liquid mating assays. They showed that the presence of MWCNTs could promote the transfer of plasmid. The authors concluded that the agglomeration of MWCNTs increased the plasmid transfer rates. The effects of reactive oxygen species (ROS) caused by the presence of MWCNTs might not be large for the plasmid transfer. This study contained interesting aspects, but some parts of the manuscript was not necessarily clear to this reviewer.

We thank the reviewer for their positive assessment of our study and have improved the manuscript according to the reviewer's comments.

2. They used IncP-1epsilon plasmid, pJK5 as a model plasmid. This reviewer could understand that the authors always used this plasmid as a model, but was the promotion of plasmid transfer commonly observed in other plasmids? Did the authors have any comments on it?

We agree with the reviewer that testing if the observed dynamics hold true when using a different plasmid or donor strain is valuable. We have hence repeated the basic mating experiment using *E. coli* instead of *P. putida* as a donor and plasmid pB10 instead of pJK5. We now show these additional results in the new Figure 3 and have added the following paragraphs to the manuscript:

Material and methods section:

"To confirm if observed results are exclusive to the above donor-recipient combination or to the pJK5 plasmid, experiments were repeated using *Escherichia coli* MG1655::lacIq-pLpp-mCherry-Km^R (1) as the donor strain of IncP-1β broad-host range plasmid pB10 (2, 3) that confers resistance to amoxicillin, tetracycline, sulfonamides, and streptomycin. Conditions were exactly as described above with the lone exception that the incubation temperature was increased to 37 °C."

1. Klümper U, Dechesne A, Smets BF. 2014. Protocol for Evaluating the Permissiveness of Bacterial Communities Toward Conjugal Plasmids by Quantification and Isolation of Transconjugants, p. 275–288. In *Hydrocarbon and Lipid Microbiology Protocols*, Springer Protocols Handbook. Humana Press.
2. Schlüter A, Heuer H, Szczepanowski R, Forney LJ, Thomas CM, Pühler A, Top EM. 2003. The 64 508 bp IncP-1 β antibiotic multiresistance plasmid pB10 isolated from a waste-water treatment plant provides evidence for recombination between members of different branches of the IncP-1 β group. *Microbiology* 149:3139–3153.
3. Mishra S, Klümper U, Voolaid V, Berendonk TU, Kneis D. 2021. Simultaneous estimation of parameters governing the vertical and horizontal transfer of antibiotic resistance genes. *Sci Total Environ* 798:149174.

Results section:

"Similar results were observed when using plasmid pB10, where exposure to MWCNTs at any concentration had equally no significant effect on bacterial densities of either, the *E. coli* donor or the *P. putida* recipient (all $p > 0.05$, $n = 3$, ANOVA, Figure 3)."

Figure 3: Final concentrations of bacterial donor *E. coli* with plasmid pB10 (red), recipient strain *P. putida* (grey) transconjugants (green) and ratio of transconjugants per recipient (green) after incubation with varying concentrations of MWCNTs (n = 3, 3, 3, 3). Center lines show the median; box limits indicate the 25th and 75th percentiles; whiskers extend 1.5 times the interquartile range from the 25th and 75th percentiles; outliers are represented by dots; data points are plotted as circles.

“To confirm that these trends hold true for different plasmid:donor combinations, experiments were repeated with plasmid pB10 introduced through an *E. coli* donor strain. Again the absolute number of transconjugants receiving plasmid pB10 increased significantly with increasing concentrations of MWCNTs ($r_s = 0.82048$, $p = 0.001$ of Spearman's rho being zero; Figure 3) from $1.37 \pm 0.55 * 10^6 \text{ mL}^{-1}$ in the absence of MWCNTs to $8.03 \pm 0.98 * 10^6 \text{ mL}^{-1}$ at 10 mg L^{-1} MWCNTs. Equally the percentage of transconjugants per recipient (T/R) significantly increased from $0.08 \pm 0.05 \%$ from 0 mg L^{-1} MWCNTs to $0.43 \pm 0.15 \%$ at 10 mg L^{-1} MWCNTs ($t = -3.966$, $p = 0.017$, $n = 3$, t-test, Figure 3). As the observed dynamics were nearly identical for the different donor:plasmid combinations, subsequent analysis regarding the underlying mechanism was carried out exclusively for the *P. putida* donor and plasmid pKJK5.”

Discussion section:

“This effect was demonstrated for two different plasmid:donor combinations.”

3. As for the ROS, did they compare the amount of ROS in the absence and presence of MWCNT?

In our experiments using the ROS scavenger thiourea no effect of ROS on the plasmid transfer rates was observed. Consequently, we did not measure the exact amount of ROS in the two treatments. While we agree with the reviewer that the exact amount of ROS between the treatments would in general be interesting to examine, we deemed this not essential to the story of the manuscript as no ROS effect was observed.

4. Figure 3. The y-axis of the left graph should be shown in exponential representation. Did the authors

have any comments that the CFU of transconjugants and rate of transconjugant decreased in the presence of 0.1 mg/L MWCNT?

The axis has been adjusted, you can find it in the bottom right of Figure 2, as the original Figure 2 and 3 have been combined in this revised version.

In addition, we would like to point out that it might seem by visual inspection like the transconjugants and rate of transconjugants decreased at the lowest concentration. However, statistical evaluation of the data revealed that there is no significant difference between the 0.1 mg/L MWCNT treatment and the non-MWCNT control, hence it is not explicitly discussed in the manuscript.

5. Lines 159-190, 236-247 and Figure 4. This reviewer could not understand the meaning why they introduced the modeling of plasmid transfer on MWCNT exposure. It might be helpful if the authors described why they calculated the rates, what they expected, and what the conclusion was based on the comparisons between the modeling and observations.

In fact, the qualitative impact of MWCNT exposure on plasmid transfer can already be inferred from empirical measures such as T/R ratios. While such ratios are cheap to compute, their use is actually discouraged in favor of rate constants of a mathematical model which capture the main features of mating experiments, including initial densities, growth, and conjugation (DOI: 10.1016/j.jtbi.2011.10.034). Moreover, model fitting allows the impact of MWCNT on plasmid transfer to be described not only qualitatively but also quantitatively (e.g. a direct concentration dependency) in a consistent framework. We have now included this reasoning in the manuscript.

Material and methods section:

“Here, we amend the equations of Levin et al. (4) with an empirical term reflecting the effect of MWCNT on the rate constant of plasmid transfer. In contrast to empirical measures of plasmid transfer efficiency like, e.g., T/R ratios, such model-based rate constants are more specific and better suited for comparison across different studies (5).”

4. *Levin BR, Stewart FM, Rice VA. 1979. The kinetics of conjugative plasmid transmission: Fit of a simple mass action model. Plasmid 2:247–260.*

5. *Zhong X, Droesch J, Fox R, Top EM, Krone SM. 2012. On the meaning and estimation of plasmid transfer rates for surface-associated and well-mixed bacterial populations. J Theor Biol 294:144–152.*

Results section:

“To determine the nature of the relationship between MWCNT exposure and plasmid transfer (e.g. a direct concentration dependence) we employed modelling of the plasmid transfer dynamics of plasmid pKJK5 between the *P. putida* donor and recipient.”

6. Line 253-253. The data were not shown. Please describe more clearly here, i.e., "(data not shown)".

We have added (data not shown) accordingly.

7. Lines 275-280 and Figure 6. They just showed one figure here, and the figure might be a

representative one in the microscopy images. Was it difficult to show more examples? How frequently did the observed such images?

Around 200 particles per mL of the same size as shown in the Figure can be observed based on the particle size distribution measured for the highest concentration of MWCNTs. These numbers have further been corroborated through microscopic counts. The reviewer is right that to show that the observations are indeed general we should include additional images of similar nature into the figure. We consequently added 2 additional microscopic images of MWCNT agglomerates with attached bacteria to Figure 6.

8. Lines 315-316. It was not necessarily clear why the authors considered the naked plasmid DNA in their experiments were negligibly low. It might be better to show the amount of naked plasmid DNA in the past report in ref. #44.

We agree with this comment by the reviewer that a comparison with the previous study is beneficial to clarify our statements. It now reads:

“However, in this previous study high concentrations ($0.3 \mu\text{g mL}^{-1}$) of specifically extracted plasmid DNA were necessary to observe effects. Contrary in our experimental design, as well as in most environmental settings, concentrations of naked plasmid DNA, if present due to bacterial lysis in the culture, would be expected at several orders of magnitude lower. Furthermore, the association of this free DNA with the here used, non-functionalized MWCNTs is highly unlikely.”

March 21, 2022

Dr. Uli Klümper
TU Dresden
Zellescher Weg
dresden
Germany

Re: Spectrum00410-22R1 (Multi-walled carbon nanotubes promote bacterial conjugative plasmid transfer)

Dear Dr. Uli Klümper:

Your manuscript has been accepted, and I am forwarding it to the ASM Journals Department for publication. You will be notified when your proofs are ready to be viewed.

Sincerely,

Zhenjiang Xu
Editor, Microbiology Spectrum
